# Ventral hippocampus to nucleus accumbens shell circuit regulates approach decisions during motivational conflict

Dylan Patterson[1], Nisma Khan[2], Emily A. Collins[2], Norman R. Stewart[1], Kian Sassaninejad[2], Dylan Yeates[2], Andy C. H. Lee[2,3], Rutsuko Ito[1,2]*

1 Department of Cell and Systems Biology, University of Toronto, Toronto, Canada, 2 Department of Psychology (Scarborough), University of Toronto, Toronto, Canada, 3 Rotman Research Institute, Baycrest Centre, Toronto, Canada

* rutsuko.ito@utoronto.ca

**Data Availability Statement:** All data files are available from the Open Science Framework database under the https://osf.io/pwcer/.

## Abstract

Successful resolution of approach-avoidance conflict (AAC) is fundamentally important for survival, and its dysregulation is a hallmark of many neuropsychiatric disorders, and yet the underlying neural circuit mechanisms are not well elucidated. Converging human and animal research has implicated the anterior/ventral hippocampus (vHPC) as a key node in arbitrating AAC in a region-specific manner. In this study, we sought to target the vHPC CA1 projection pathway to the nucleus accumbens (NAc) to delineate its contribution to AAC decision-making, particularly in the arbitration of learned reward and punishment signals, as well as innate signals. To this end, we used pathway-specific chemogenetics in male and female Long Evans rats to inhibit the NAc shell projecting vHPC CA1 neurons while rats underwent a test in which cues of positive and negative valence were presented concurrently to elicit AAC. Additional behavioral assays of social preference and memory, reward and punishment cue processing, anxiety, and novelty processing were administered to further interrogate the conditions under which the vCA1-NAc shell pathway is recruited. Chemogenetic inhibition of the vCA1-NAc shell circuit resulted in animals exhibiting increased decision-making time and avoidance bias specifically in the face of motivational conflict, as the same behavioral phenotype was absent in separate conditioned cue preference and avoidance tests. vCA1-NAc shell inhibition also led to a reduction in seeking social interaction with a novel rat but did not alter anxiety-like behaviors. The vCA1-NAc shell circuit is therefore critically engaged in biasing decisions to approach in the face of social novelty and approach-avoidance conflict. Dysregulation of this circuit could lead to the precipitation of addictive behaviors in substance abuse, or maladaptive avoidance in situations of approach-avoidance conflict.

## Introduction

Approach-avoidance conflict (AAC) resolution is a form of decision-making that requires the effective evaluation of stimuli with mixed (positive and negative) valence. These opposing

**Funding:** This study was funded by a project grant from the Canadian Institutes of Health Research awarded to RI & ACHL (156070), Canada Graduate Scholarship awarded to DP by Social Sciences and Humanities Research Council (SSHRC), Canada Graduate Scholarship awarded to EC by Canadian Institutes of Health Research, Canada Graduate Scholarship awarded to NK and NS by Natural Sciences and Engineering Research Council of Canada. The funders had no role in study design, data collection and analysis, decision to publish, or preparation of the manuscript.

**Competing interests:** The authors have declared that no competing interests exist.

**Abbreviations:** AAC, approach-avoidance conflict; CCA, conditioned cue avoidance; CCP, conditioned cue preference; CNO, clozapine N-oxide; EPM, elevated plus maze; HPC, hippocampus; IC, intra-cerebral; LS, lateral septum; NAc, nucleus accumbens; vHPC, ventral hippocampus.

associations simultaneously incentivize an organism to approach and avoid the stimulus, creating a motivational conflict [1,2]. The successful resolution of this conflict is necessary for survival and adaptive daily living, for instance, in situations where an organism must forage for food under threat of predation or decide whether to indulge in a high calorie treat that may lead to straying from dietary goals. Importantly, it is hypothesized that AAC resolution is disrupted in mental health disorders, with maladaptive approach biases linked to substance abuse patients and preclinical models of addictive behaviors, and avoidance biases associated with individuals with anxiety and depression [3–7]. These findings underscore the importance of elucidating the neural circuitry that supports approach-avoidance decision-making.

A network of structures including the prefrontal cortex, striatum, and limbic regions such as the hippocampus (HPC), amygdala, thalamus, and hypothalamus has been implicated in mediating decision-making under AAC [8–14]. In particular, recent convergent human and animal research has identified the ventral aspect of the HPC (vHPC, anterior in humans) as a key node in the regulation of AAC resolution, driving approach and avoidance behaviors in a subfield-specific manner [15,16]. Schumacher and colleagues found that GABAR-mediated inactivation of the ventral CA1 (vCA1) subfield increased avoidance tendencies, while inactivation of the ventral CA3 (vCA3) subfield increased approach tendencies in the face of conflict, thus implicating the vCA1 in mediating cued approach behavior and the vCA3 in cued avoidance behavior [15]. The divergent functions of these vHPC subregions in approach-avoidance conflict processing raises the question of whether this difference is mediated by their unique connections to other brain areas. To address this, we have recently identified the extrinsic connectivity of the vCA3 with the lateral septum, the sole target of long range vCA3 projections, to be critical in mediating cued avoidance behavior under conflict [17].

The vCA1, unlike the vCA3, has extensive connectivity downstream [18], among which, the strong connectivity between the vCA1 and nucleus accumbens (NAc) is of particular interest in modulating approach-avoidance behaviors in the face of motivational conflict [19]. Traditionally, the NAc has been thought to mediate reward-directed behavior and act as a limbic-motor interface that functionally links motivation and action [20]. However, more recent evidence suggests the NAc is involved in the evaluation and integration of both positively and negatively valenced information and their influence over behavioral output in a regionally specific manner [21–26]. For instance, NAc shell GABA microcircuits are thought to represent appetitive to aversive information along a rostro-caudal gradient [17–18]. The NAc core is implicated in mediating cued influences over pavlovian or goal-directed behavior [27], and GABA-mediated inhibition of the caudal core has also been shown to induce an avoidance bias during exposure to a cue inducing AAC [14].

Glutamatergic projections from the vCA1 to the NAc predominantly target the shell, and this pathway has been implicated in promoting stress-induced negative affective/motivational states [28], inhibiting feeding and reward-seeking [29,30], as well as promoting positively motivated behaviors such as social [31,32] and contextual [33] reward-seeking. However, the role of NAc shell-projecting vCA1 in guiding approach-avoidance choices when faced with stimuli of opposing valence concurrently remains unexplored. To address this, animals underwent chemogenetic inhibition of the vCA1-shell pathway while performing a cued conflict test in which they were given a choice to explore a learned cue imbued with both appetitive and aversive qualities (to induce conflict), and a cue associated with no outcome. Animals were also assessed in other forms of AAC using additional tests, including the sociability and social memory test, where they faced the choice of interacting with a novel conspecific—which may elicit an AAC—and ethological anxiety tests (elevated plus maze and open field) that elicit a safety versus exploration conflict. We report that inhibiting vCA1 glutamatergic afferents to the NAc shell disrupts decisions to approach specifically under situations of AAC that evoke

opposing motivations (positive and negative), revealing a hitherto undiscovered role of vCA1--NAc shell in approach-avoidance conflict resolution.

## Results

### Histological verification confirms vCA1-NAc circuit targeting, with reduced c-Fos+ cells in the NAc shell following CNO administration

Histological verification of viral expression and injector placement revealed viral transduction within the vCA1 (and sometimes vCA3) regions of the vHPC for both hM4Di-mCherry and GFP constructs. Terminal expression of the constructs was found to be confined to the NAc shell and boundary between the shell and core, with cannula and infuser tip placement verified to be located within the NAc shell to enable targeted inhibition of NAc-projecting vCA1 neurons (Fig 1A–1C). Assessment of c-Fos positive (+) cell density in the NAc shell and core areas revealed that microinfusion of CNO induced a significant decrease in c-Fos+ cells in the NAc shell ($p < 0.001$), but not in the adjacent core area ($p = 0.54$) of hM4Di-mCherry expressing animals, compared to those microinfused with saline (Fig 1D, F$region \times drug$ (1,29) = 14.84, $p < 0.001$, $\eta^2 = 0.34$), thereby verifying the dampening of activity of NAc shell-projecting vCA1 neurons. The density of c-Fos+ cells was lower overall in the females than males (F$sex$ (1,29) = 4.70, $p = 0.038$, $\eta^2 = 0.14$), with no significant sex differences observed based on drug treatment (F$sex \times drug$ (1,29) = 2.11, $p = 0.16$). The microinjection of CNO combined with fluorescent rhodamine dye in a small subset of GFP-expressing animals showed that CNO diffusion was contained within the targeted GFP-expressing vHPC terminals in the NAc, with minimum spreading to overlying areas (which also receive vHPC projections) such as the lateral septum (S1 Fig). However, it is acknowledged that, despite the efforts to confirm localized diffusion presented herein, a limitation of this approach is the possibility of CNO spreading into adjacent or overlying regions due to individual variability; albeit it should be noted that previous work from our laboratory has demonstrated that chemogenetic inhibition of vCA1 terminals in the lateral septum produces effects opposite to those observed here on AAC [17].

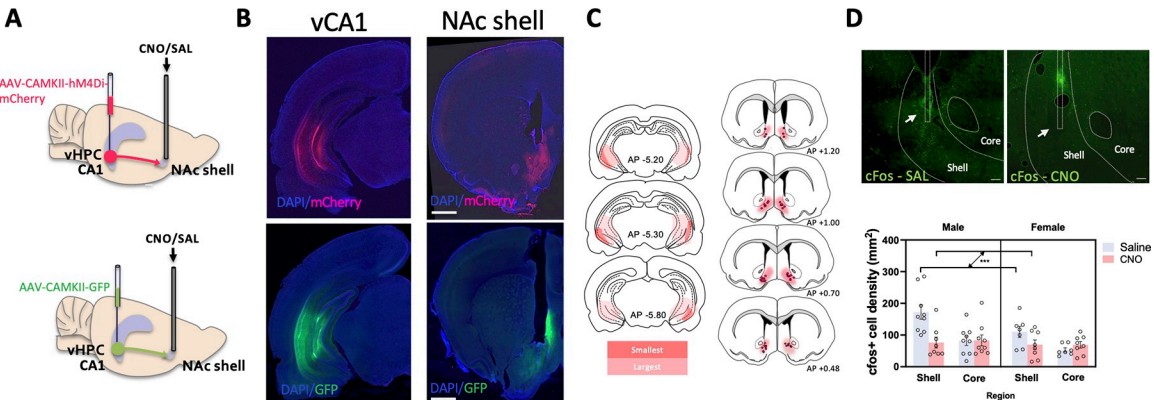

**Fig 1. Chemogenetic inhibition of ventral hippocampus CA1 (vCA1) to NAc shell projections.** (**A**) Schematic diagram of inhibitory DREADDS (hM4Di-mCherry) or empty vector (GFP) transduction in vCA1 and cannula implantation in NAc shell via which CNO or saline was microinfused for pathway-specific inhibition. (**B**) Representative images of somatic transduction of hM4Di-mCherry or empty vector GFP in vCA1 (left) and axonal transport of the viral load in the NAc shell (right). Scale bar, 1 mm. (**C**) Cannula placements in the NAc shell (DREADDs group only) and the minimum and maximum DREADDs expression in vCA1 and NAc shell, AP: Anterior-Posterior to bregma. (**D**) Representative images of c-Fos expression in the NAc following either saline (SAL) or CNO (1 mM) infusion, showing cannula tract, and tip localized within the shell subregion (arrow). Scale bar, 100 μm. Intracerebral CNO infusions attenuated c-Fos signal in areas of the NAc shell that received hM4Di positive inputs (bottom graph). c-Fos-positive cell density in the adjacent NAc core was not affected by CNO infusions. Data show means ± SEM. ***$p < 0.001$. The data underlying panel D are available at: https://osf.io/pwcer/. CNO, clozapine N-oxide; NAc, nucleus accumbens.

## vCA1-NAc shell inhibition induced a decision bias to avoid during approach-avoidance conflict

All animals were first assigned to one of 4 groups (hM4Di-SAL, hM4Di-CNO, GFP-SAL, and GFP-CNO) to undergo a dual surgery. This included microinfusion of an AAV containing inhibitory DREADDs (hM4Di) or an empty vector (GFP control) into the vCA1, as well as implantation of a guide cannula into the vCA1 terminals in the NAc for later infusions of CNO (for targeted vCA1-NAc circuit inhibition) or saline (control).

Animals were then trained to associate 3 visual-tactile cues with distinct outcomes: appetitive, neutral, and aversive, in the radial maze (Fig 2A). Every 4 conditioning sessions (total 8 to 12 sessions), learning was assessed using a conditioned cue acquisition test in which animals were allowed to freely explore the cues for 5 min in the absence of any reinforcement. All but 2 animals successfully acquired cue valence by their final (second or third) acquisition test (Fig 2B, hM4Di: F$cue$(1.62, 48.49) = 124.43, $\eta^2$ = 0.81; $p$ < 0.001, S2A Fig, GFP: F$cue$(1.47, 42.75) = 62.47, $p$ < 0.001, $\eta^2$ = 0.68) and spent significantly more time exploring the appetitive arm compared to the neutral arm ($p$ < 0.001) and significantly less time exploring the aversive arm compared to the neutral arm ($p$ < 0.001). There were no significant differences in the cue acquisition pattern between sex (male versus female) and future drug (saline versus CNO) groups (hM4Di: F$sex$ x $drug$ x $cue$ (2,60) = 0.13; $p$ = 0.88, GFP: F$sex$ x $drug$ x $cue$ (2,58) = 0.26; $p$ = 0.77, all other interactions $p$ > 0.13).

Following successful cue acquisition, animals underwent a conflict test, in which they were given a choice to explore an arm with the neutral cue and an arm with both the appetitive and aversive cues superimposed (conflict arm) under extinction conditions. Chemogenetic inhibition of the vCA1-NAc shell pathway significantly reduced the time spent exploring the conflict and neutral cues compared to the saline-infused hM4Di group (Fig 2C, F$drug$ (1,30) = 5.25, $p$ = 0.029, $\eta^2$ = 0.15, F$cue$ x $drug$ (1,30) = 3.62, $p$ = 0.067, $\eta^2$ = 0.11), irrespective of sex (F$drug$ x $sex$ (1,30) = 0.33, $p$ = 0.64). A separate analysis of the difference score (time spent in conflict arm minus time spent in neutral arm) revealed that CNO-infused hM4Di rats spent less time in the conflict arm relative to the neutral arm compared to saline-infused controls (Fig 2D, Difference score, t(32) = 2.03, $p$ = 0.05). An additional assessment of the central hub time revealed that chemogenetic vCA1-NAc shell inhibition caused a significant increase in the time spent within the hub compared to hM4Di saline group (Fig 2E; F$drug$ (1,30) = 6.26 $p$ = 0.018, $\eta^2$ = 0.17). No significant differences between the times spent exploring the conflict, neutral arms, and hub were observed between saline- and CNO-infused GFP animals (S2B–S2D Fig, Time spent: F$drug$(1,29) = 3.04, $p$ = 0.09, F$cue$(1,29) = 1.57, $p$ = 0.22, Difference Score: t(31) = 0.03, $p$ = 0.97, Hub time: F$drug$(1,29) = 1.41, $p$ = 0.24).

To better understand the factors contributing to the increased hub time in the CNO-infused hM4Di group, we also analyzed risk-assessment behavior, which was defined as the rats "checking" back and forth between the 2 arms before committing to their first entry into an arm. CNO-infused hM4Di animals exhibited significantly more "checking" behavior than the saline controls prior to their first arm entry (Fig 2F, F$drug$(1,30) = 6.26; $p$ = 0.018, $\eta^2$ = 0.17), irrespective of sex (F$sex$ (1.30) = 0.08, $p$ = 0.78). The latencies to enter into the conflict and neutral arms for the first time were also analyzed, and we found that CNO-infused hM4Di males, but not females, were significantly slower to make entries into the conflict arm compared to saline-infused hM4Di male controls ($p$ = 0.029), as well as their female CNO-infused hM4Di counterparts (Fig 2G and 2H, $p$ = 0.023, F$sex$ x $drug$ x $cue$ (1, 30) = 4.29; $p$ = 0.047, $\eta^2$ = 0.13). CNO infusion in GFP-expressing animals failed to elicit any changes in risk assessment behavior (S2E Fig, F$drug$(1,29) = 0.19, $p$ = 0.67, F$drug$ x $sex$(1,29) = 0.11, $p$ = 0.75) and latencies to enter the conflict and neutral arms for the first time (S2F and S2G Fig, F$drug$ (1,29) = 0.57, $p$ = 0.46, all interactions with drug, $p$ > 0.47, with sex, $p$ > 0.17).

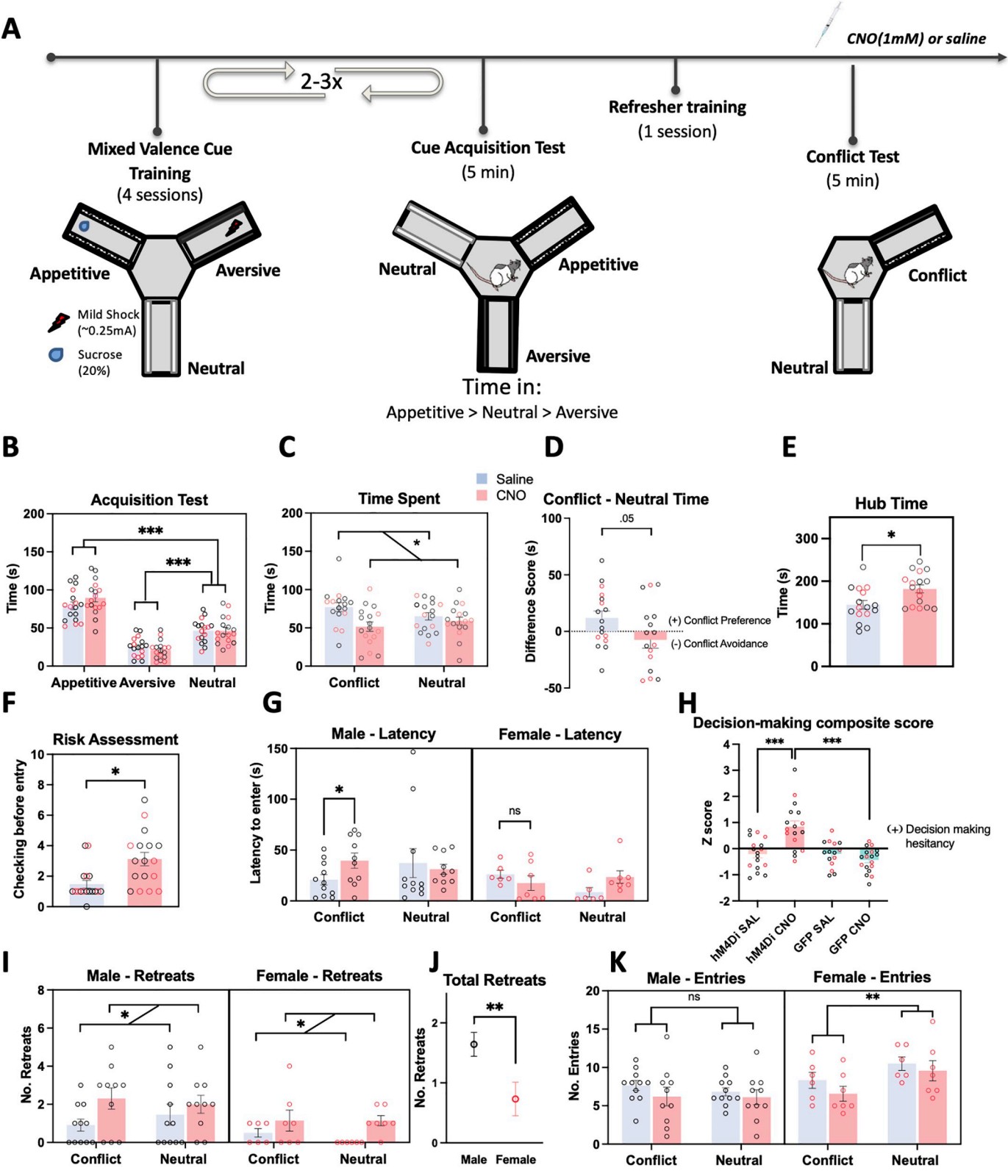

**Fig 2. Effect of chemogenetic inhibition of vCA1-NAc shell on approach-avoidance conflict.** (**A**) Experimental timeline showing mixed valence cue training (2 or 3 × 4 sessions), followed by a cue acquisition test (5 min), refresher training session, and finally the cue conflict test, prior to which animals received intracranial microinfusion of CNO (1 mM) or saline. (**B**) After 2 to 3 rounds of 4 conditioning sessions, 34 hM4Di-expressing animals (hM4Di Sal: $n$ = 17 (6 females), hM4Di CNO: $n$ = 17 (7 females), symbols with red border depict female data) successfully acquired cue-outcome associations, demonstrating a greater amount of time in the appetitive vs. the neutral arm ($p < 0.001$) and less time in the aversive vs. the neutral arm ($p < 0.001$). (**C**) In the conflict test, rats were given a choice to explore a neutral cued arm or an arm with the appetitive and aversive cues superimposed (conflict cue). Time spent in the conflict and neutral cued arm was significantly reduced in the vCA1-NAc shell inhibited group (hM4Di CNO) compared to control groups (hM4Di SAL). (**D**) CNO-infused hM4Di animals avoided the conflict cued arm relative to the control hM4Di SAL animals. (**E**) Amount of time spent in the hub (central compartment), and (**F**) risk assessment (checking) behavior were also significantly increased in the vCA1-NAc shell inhibited group, compared to the control SAL group. (**G**) Latency to enter the conflict cue arms for the first time was increased in the males but not females after vCA1-NAc shell inhibition. (**H**) When a decision-making composite z score was computed and compared across the 4 groups (including GFP-expressing CNO and SAL groups), the hM4Di-CNO group exhibited significantly higher scores, indicative of potentiated decision-making hesitancy. (**I**) Number of retreats in both conflict and neutral arms were more pronounced in the vCA1-NAc shell inhibited group compared to the control group, and (**J**) the males exhibited more retreat behavior overall than females. (**K**) vCA1-NAc shell inhibition did not alter arm entries; however, the females made more entries into the neutral than conflict arms. Data show means ± SEM. * $p < 0.01$, ** $p < 0.01$, ***$p < 0.001$. The data underlying this figure are available at: https://osf.io/pwcer/. CNO, clozapine N-oxide; NAc, nucleus accumbens.

Given that vCA1-shell inhibition increased hub time, "checking" behavior, and latency to enter the conflict arm—all of which can be considered indices of prolonged decision-making —we computed a decision-making composite score. Specifically, we z-scored each of the 3 measures across sex, drug, and virus groups and then averaged the resulting z-scores to obtain a single composite score. Analysis of the composite z score confirmed that CNO-infused hM4Di animals exhibited higher decision-making composite z-scores compared to all other groups (hM4Di CNO versus SAL: $p = 0.001$, HM4Di CNO versus GFP CNO: $p < 0.001$), indicative of increased indecision in the face of motivational conflict, irrespective of sex (Fig 2I, F$drug$ $x$ $virus$ (1,59) = 8.19, $p = 0.006$, $\eta^2 = 0.12$; F$sex$ $x$ $drug$ $x$ $virus$ (1,59) = 3.69, $p = 0.06$).

In addition, we analyzed the number of "retreats" (back-treading) exhibited by the rats once they entered the arms as an index of aversion. CNO-infused hM4Di animals exhibited significantly more retreats overall (irrespective of cue type), compared to saline-infused hM4Di controls (Fig 2J and 2K, F$drug$ (1, 30) = 7.44; $p = 0.011$, $\eta^2 = 0.20$, F$drug$ $x$ $cue$ (1, 30) = 0.67; $p = 0.80$). Additionally, hM4Di males showed significantly more retreats overall than the females (F$sex$ (1, 30) = 8.08; $p = 0.01$, $\eta^2 = 0.21$). However, the same sex effect was not observed in the GFP group (F$sex$ (1, 29) = 0.41; $p = 0.53$) suggesting that the effect in the hM4Di groups were largely driven by the potentiated retreats in the CNO-infused hM4Di males. Finally, vCA1-NAc circuit inhibition did not induce any significant changes in the number of entries into the conflict and neutral arms (Fig 2L and 2M, F$drug$(1, 30) = 1.99; $p = 0.17$, all interactions involving drug, $p > 0.43$). The female hM4Di rats entered the neutral-cued arm more frequently than the conflict-cued arm relative to male hM4Di rats (F$sex$ $x$ $cue$(1, 30) = 9.94; $p = 0.004$, $\eta^2 = 0.25$), albeit the same effect was not observed in GFP-expressing rats (F$sex$ $x$ $cue$ (1, 29) = 0.59; $p = 0.45$). CNO infusion in GFP rats did not induce any alteration in retreats (S2H and S2I Fig, F$drug$ (1, 29) = 0.03; $p = 0.87$), or entries into the cued arms (S2J and S2K Fig, F$drug$ (1, 29) = 0.30; $p = 0.59$).

## vCA1-NAc shell inhibition did not affect approach-avoidance decisions in non-conflict situations

Next, to assess if vCA1-NAc shell inhibition influenced cued approach-avoidance decisions in non-conflict situations, conditioned cue preference (CCP) and conditioned cue avoidance (CCA) tests were conducted separately, in which the neutral arm was now presented alongside an appetitively cued (CCP), or aversively cued (CCA) arm (Fig 3A). All hM4Di animals spent significantly more time in the appetitive arm than the neutral arm in the CCP test (Fig 3B, F$cue$ (1, 28) = 43.99; $p < 0.001$, $\eta^2 = 0.61$), irrespective of drug group or sex (F$sex$ $x$ $drug$ $x$ $cue$ (1, 28) = 0.14; $p = 0.71$). All animals also spent significantly less time in the aversive arm than

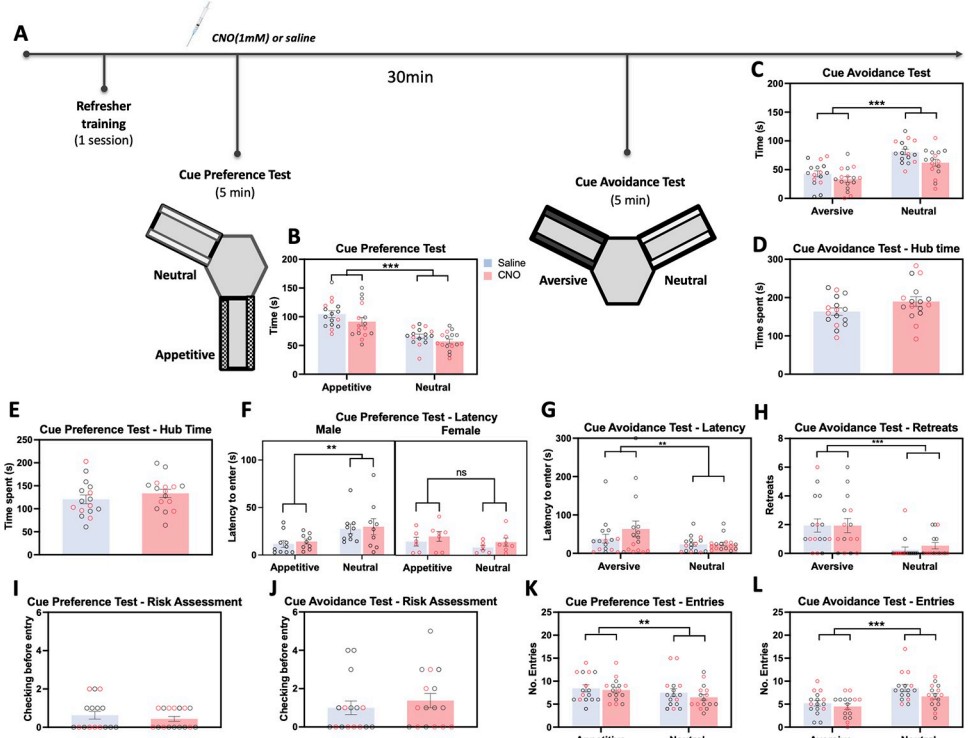

**Fig 3. No effect of vCA1-NAc shell inhibition on cue preference and avoidance tests.** (**A**) Experimental timeline showing the administration of a refresher conditioning session following the conflict test, then the administration of cue preference (appetitive vs. neutral cues) and cue avoidance (aversive vs. neutral cues) tests in counterbalanced order, prior to which animals were microinfused with CNO or saline. (**B, C**) All animals (hM4Di Sal: $n$ = 16 (6 females), hM4Di CNO: $n$ = 16 (7 females), symbols with red border depict female data) spent significantly more time in the appetitive vs. neutral arm in the cue preference test, and less time in the aversive vs. neutral arm in the cue avoidance test, irrespective of drug treatment or sex. (**D, E**) There was no group differences (drug treatment, sex) in the hub time during the cue preference test and cue avoidance tests. (**F, G**) Latency to enter the appetitive vs. neutral arms was lower in the males but not the females. Latency to enter the aversive cued arm for the first time was significantly higher compared to latency to enter the neutral arm in all groups. (**H**) The number of retreats observed in the aversively cued arm was significantly higher than in the neutral cued arms in the cue avoidance test. (**I, J**) There was no difference in risk assessment behavior between all groups in cue preference and avoidance tests. (**K, L**) All animals made more entries into the appetitive arm over the neutral arm in the cue preference test and made less entries into the aversive arm over the neutral arm in the cue avoidance test, irrespective of pathway inhibition or sex. Data show means ± SEM. * $p < 0.01$, ** $p < 0.01$, *** $p < 0.001$. The data underlying this figure are available at: https://osf.io/pwcer/. CNO, clozapine N-oxide; NAc, nucleus accumbens.

the neutral arm in the cue avoidance test regardless of drug group or sex (Fig 3C, F*cue* (1, 28) = 60.25; $p < 0.001$, $\eta^2$ = 0.68, F*sex x drug x cue* (1, 28) = 0.03; $p$ = 0.96). The time spent in the hub during CCP and CCA were not significantly different between the CNO-infused and saline-infused hM4Di rats (Fig 3D and 3E, all tests with Drug $p > 0.11$ for CCP and $p > 0.37$ for CCA). Latencies to enter the arms during CPP and CPA were not significantly different following CNO infusions in hM4Di-expressing rats (Fig 3F and 3G, all tests involving Drug, $p > 0.21$ for CCP, $p > 0.26$ for CCA), with rats taking marginally longer to enter the aversively cued arm (F*cue* (1, 28) = 4.02; $p$ = 0.055, $\eta^2$ = 0.13). In CCP, the hM4Di males showed significantly diminished first entry time into the appetitively cued arm relative to the neutral cued arm ($p$ = 0.009), but the hM4Di females failed to show this ($p$ = 0.3, F*cue x sex* (1,28) = 6.77; $p$ = 0.015, $\eta^2$ = 0.20). As expected, the number of retreats was significantly higher in the aversively cued arm versus neutral arm in all animals irrespective of drug or sex (Fig 3H, F*cue* (1, 28) = 18.31; $p < 0.001$, $\eta^2$ = 0.40, all other tests $p > 0.07$) and retreat behavior was never

observed during CPP (all 0). Risk assessment behavior was observed significantly more during CPA than CPP (Fig 3I and 3J, F$test$ (1,28) = 4.76, $p$ = 0.038, $\eta^2$ = 0.15), with no significant difference in the performance between CNO and saline-infused hM4Di (all tests with drug, $p > 0.22$). Finally, rats made more entries into the appetitively cued arm compared to neutral arm (Fig 3K, F$cue$ (1, 28) = 6.51; $p$ = 0.016, $\eta^2$ = 0.19) and less entries into the aversively cued arm (Fig 3L, F$cue$ (1, 28) = 45.95; $p < 0.001$, $\eta^2$ = 0.62) with no significant changes in entries following CNO infusions (CCP: F$drug$ (1, 28) = 1.34; $p$ = 0.26, CCA: F$drug$ (1, 28) = 3.62; $p$ = 0.067).

GFP rats showed expected preference for the appetitively cued arm in CCP, irrespective of drug or sex (S3A Fig, F$cue$ (1, 28) = 33.48; $p < 0.001$, $\eta^2$ = 0.55, F$sex \ x \ drug \ x \ cue$ (1, 28) = 0.14; $p$ = 0.71). This was accompanied by increased entries into the appetitive arm (S3B Fig, F$cue$ (1, 28) = 7.08; $p$ = 0.013, $\eta^2$ = 0.20, all drug and sex effects and interactions, $p > 0.32$), but the first entry latency into the appetitively cued arm relative to the neutral arm did not reach significance (S3C Fig, F$cue$ (1, 28) = 2.25; $p$ = 0.15, but all drug and sex effects and interactions, $p > 0.31$). CNO infusions in GFP expressing rats had no effect on hub time or risk assessment (S3D and S3E Fig, Hub; F$drug$ (1,28) = 0.17, $p$ = 0.69, Checks: F$drug$ (1,28) = 0.42, $p$ = 0.52). GFP rats showed expected avoidance of the aversively cued arm in CCA (S3F Fig, F$cue$ (1, 28) = 54.85; $p < 0.001$, $\eta^2$ = 0.66). However, there was a significant interaction between sex, drug, and cue type (F$sex \ x \ drug \ x \ cue$ (1, 28) = 6.28; $p$ = 0.018, $\eta^2$ = 0.18). Post hoc comparisons revealed that none of the drug effects were significant (clozapine N-oxide (CNO) versus SAL, $p > 0.11$), and no significant sex differences were observed (all $p > 0.064$). Furthermore, the times spent in the aversive versus neutral cued arms were significantly different, with the $p$-values ranging from 0.03 to <0.001. Other indices of avoidance of the aversive arm confirmed these results. First entry latency into the aversively cued arm relative to the neutral arm was significantly lower (S3G Fig, F$cue$ (1, 28) = 7.74; $p$ = 0.01, $\eta^2$ = 0.22, all drug and sex effects and interactions, $p > 0.08$), and animals exhibited decreased entries into the aversive arm (S3H Fig, F$cue$ (1, 28) = 25.01; $p < 0.001$, $\eta^2$ = 0.47, all drug and sex effects and interactions, $p > 0.062$). Animals also showed increased retreat behaviors in the aversive versus neutral arms (S3I Fig, F$cue$ (1, 28) = 24.36; $p < 0.001$, $\eta^2$ = 0.47, all drug and sex effects and interactions, $p > 0.25$). Crucially, CNO infusions in GFP expressing rats had no effect on hub time or risk assessment in CCA (S3J and S3K Fig, Hub; F$drug$ (1,28) = 0.17, $p$ = 0.69, Checks: F$drug$ (1,28) = 0.42, $p$ = 0.52).

Altogether, these results indicate that chemogenetic inhibition of the vCA1-NAc shell pathway did not affect conditioned cue preference or avoidance behavior per se, and that the observed effects of vCA1-NAc shell inhibition were specific to the conflict test.

## vCA1-NAc shell inhibition did not affect approach-avoidance decisions towards novel bar cues

We also conducted a cue novelty preference test (Fig 4A) in a subset of animals, in which a novel set of cues was presented in one arm of the Y-maze, and the neutral cues in another arm to assess whether the observed effects in the conflict test were a function of the relative novelty of the conflict cue. Novelty cue preference was evident in both CNO injected hM4Di and GFP groups, with animals spending more time exploring the novel cues over the neutral cues (Fig 4B, F$arm$ (1,14) = 5.01; $p$ = 0.042, $\eta^2$ = 0.26, no significant effects of virus (all $p > 0.07$) or sex (all $p > 0.42$)). No significant differences were observed between the hM4Di and GFP groups in hub time (Fig 4C, F$virus$ (1,14) = 0.054; $p$ = 0.82), risk assessment "checks" (Fig 4D, F$virus$ (1,14) = 2.91, $p$ = 0.11), or the latency to enter either of the arms (Fig 4E, F$virus$ (1,14) = 2.50, $p$ = 0.14), albeit there was an effect reaching significance in the total number of retreats, with

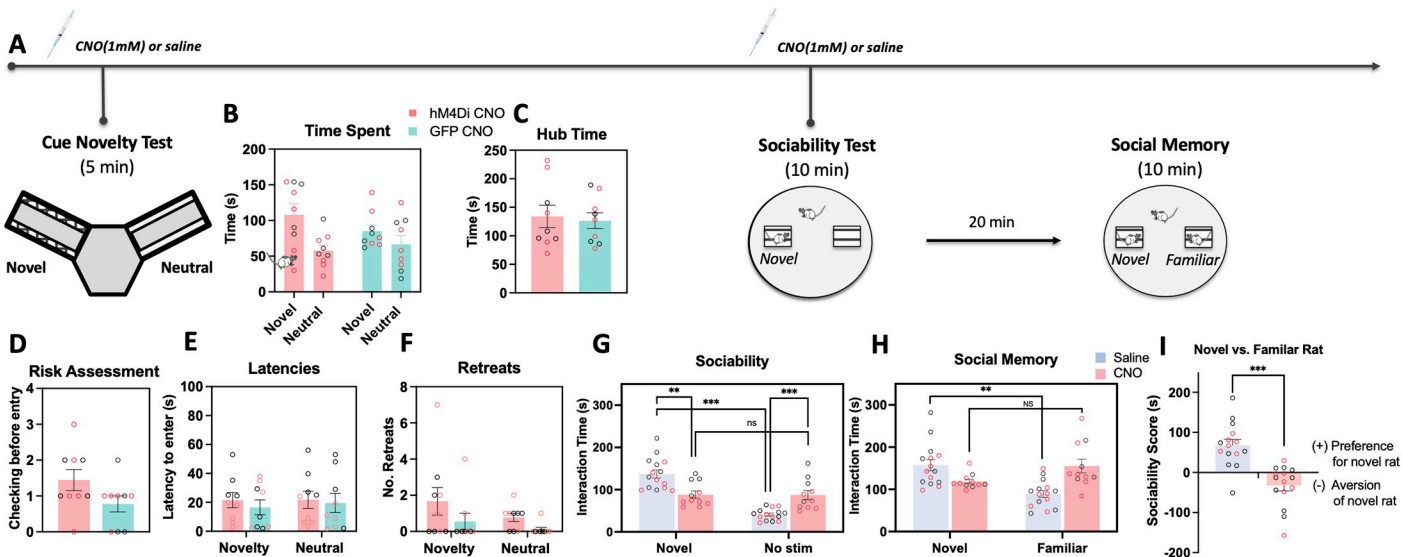

**Fig 4. Effect of vCA1-NAc shell inhibition on assays of novelty processing, social seeking, and discrimination.** (**A**) Timeline of experimental procedures; following the completion of the cue preference/avoidance tests, animals underwent testing in: cue novelty test, social interaction and social novelty seeking tests. Prior to each of the tests, animals were microinfused with CNO or saline. (**B**) Animals (hM4DI CNO: $n = 9$ (5 females), GFP CNO: $n = 9$ (5 females), symbols with red border depict female data) spent more time engaging with novel over neutral cues in the Y-maze, regardless of drug treatment or sex. No significant drug treatment or sex differences were observed with (**C**) hub time, (**D**) risk assessment, (**E**) latencies to first entry into the novel vs. neutral cued arms, and (**F**) retreats in the arms. (**G**) vCA1-NAc shell inhibited animals spent significantly less time interacting with the novel conspecific over an empty cage compared to SAL control group and failed to show preference for social interaction (hM4Di Sal: $n = 15$ (7 females), hM4Di CNO: $n = 11$ (7 females)). (**H, I**) vCA1-NAc shell inhibited animals spent more time interacting with the familiar rat and significantly less time interacting with (i.e., avoidance of) the novel rat compared to SAL controls. Data show means ± SEM. * $p < 0.01$, ** $p < 0.01$, ***$p < 0.001$, ## $p < 0.01$. The data underlying this figure are available at: https://osf.io/pwcer/. CNO, clozapine N-oxide; NAc, nucleus accumbens.

the hM4Di group showing more retreats overall than the GFP group (Fig 4F, F*virus* (1,10) = 4.24, $p = 0.060$).

## vCA1-NAc shell inhibition induced avoidance of social interaction and new conspecific

Next, to assess the role of the vCA1-NAc shell in processing of novel social stimuli that may elicit an approach/avoidance conflict, a 2 phase social discrimination task was conducted whereby animals were first given the opportunity to interact with a novel caged conspecific versus an empty cage (sociability test), and then subsequently given the choice to interact with the familiar caged conspecific from Phase 1, or a novel caged rat (social memory test). Chemo-genetic inhibition of the vCA1-NAc shell pathway significantly altered social-seeking behavior in the sociability test (Fig 4G, F*drug x stimulus* (1, 22) = 43.09; $p < 0.001$, $\eta^2 = 0.66$), with CNO-infused hM4Di animals spending significantly less time interacting with the novel caged conspecific compared to saline-infused hM4Di group ($p = 0.002$), and significantly increased time interacting with an empty cage compared to controls ($p < 0.001$). Furthermore, whereas the saline-infused control group exhibited significantly higher exploration of the caged con-specific compared to the empty cage (no social stimulus, all $p < 0.001$), vCA1-NAc shell inhib-ited animals failed to show this discrimination ($p = 0.97$). GFP-expressing rats exhibited the expected preference for social interaction (S4A Fig, F*stim* (1, 22) = 196.95; $p < 0.001$, $\eta^2 = 0.90$), with no significant effects of drug or sex (Time spent: all effects and interactions involv-ing drug and sex: $p > 0.08$).

vCA1-NAc shell pathway inhibition significantly altered social memory test performance (Fig 4H, F *drug x stim* (1, 22) = 13.95; $p < 0.001$, $\eta^2 = 0.39$), with CNO-infused hM4Di animals

failing to exhibit preference for the novel over familiar rat (hM4Di CNO: $p = 0.065$, hM4Di Sal: $p = 0.002$), instead showing significant avoidance of the novel rat, as indicated by their difference score ([Fig 4I], $F_{drug} (1, 22) = 43.09$; $p < 0.001$, $\eta^2 = 0.66$). CNO-infused hM4Di animals spent significantly more time with the familiar rat ($p = 0.002$) compared to saline controls. The drug (CNO) had no effect on overall interaction time during the memory test ($F_{drug} (1, 22) = 1.22$; $p = 0.28$), and no sex differences were observed (all effects and interactions, $p > 0.26$). Additionally, GFP-expressing rats exhibited the expected preference for the novel rat ([S4B Fig], $F_{stim} (1, 22) = 30.97$; $p < 0.001$, $\eta^2 = 0.59$), with no significant effects of drug or sex (Time spent: all effects and interactions involving drug and sex: $p > 0.24$, [S4C Fig], Difference Score: $F_{drug} (1, 22) = 0.03$; $p = 0.86$).

## vCA1-NAc shell inhibition did not affect general locomotor activity

To confirm that the decreased conflict cue exploration and increased latency to enter during the conflict test were not due to a general decrease in movement, a locomotor task was conducted ([S5A Fig]). Overall, there were no significant effects of CNO on locomotor activity ([S5B and S5F Fig], drug and interaction effects: hM4DI: $p > 0.12$, GFP: $p > 0.34$), with habituation (reduction) of locomotor activity over the 60 min session, as expected (hM4Di: $F_{bin}$ $(5.45,158.08) = 22.35$; $p < 0.001$, $\eta^2 = 0.44$, GFP: $F_{bin} (5.14,144.00) = 29.28$; $p < 0.001$, $\eta^2 = 0.51$). Significant sex differences were revealed with the hM4Di females (irrespective of infusion type) displaying elevated locomotor activity across the time bins compared to males ($F_{sex}$ $(1,29) = 6.51$, $p = 0.016$, $\eta^2 = 0.18$, $F_{binxsex}$ $F(5.45,158.08) = 2.39$, $p = 0.0036$, $\eta^2 = 0.076$). However, this effect was not observed in the GFP females relative to males ($F_{sex} (1,28) = 1.11$, $p = 0.75$, $F_{binxsex}$ $F(5.14,144.00) = 1,28$, $p = 0.28$).

## vCA1-NAc shell inhibition did not affect innate anxiety

Additionally, to rule out alterations in innate anxiety as an explanation for the increased latency and decreased conflict exploration, both elevated plus maze and open field tests were conducted. Animals spent significantly more time exploring the closed arms of the EPM compared to the open arms ([S5C and S5G Fig], hM4Di: $F_{arm} (1, 30) = 233.67$; $p < 0.001$, $\eta^2 = 0.89$, GFP: $F_{arm} (1, 28) = 220.23$; $p < 0.001$, $\eta^2 = 0.89$). However, CNO infusions in hM4Di-expressing animals did not produce significant effects in the EPM ([S5C and S5D Fig], Time in Open/Closed: $F_{drug \, x \, arm} (1, 30) = 1.29$; $p = 0.27$, Time in Center: $F_{drug} (1, 30) = 0.34$; $p = 0.34$) or the open field test ([S5E Fig], Centre time: $F_{drug} (1, 30) = 0.37$; $p = 0.55$). Additionally, no significant sex differences were observed in any of the measured behaviors (effects/interactions involving sex, EPM: $p > 0.18$, Open field: $p > 0.10$). No significant differences in the time spent in the open, closed, or center compartments in the EPM ([S5G and S5H Fig], $F_{drug} (1, 28) = 0.28$; $p = 0.60$, $F_{sex} (1, 28) = 0.30$; $p = 0.59$, interactions involving drug and sex: $p > 0.57$), and center time in the open field ([S5I Fig], $F_{drug} (1, 25) = 0.94$; $p = 0.34$, $F_{sex} (1, 25) = 0.83$; $p = 0.37$) were observed between drug conditions or sexes in GFP animals.

Together, our results indicate that vCA1-NAc shell inhibition resulted in decreased conflict cue exploration compared to controls. This behavior was accompanied by increased hub time, risk assessment behavior, retreat behavior, and latency to enter the conflict arm (in males), which is altogether indicative of potentiated decision bias to avoid in the face of approach-avoidance conflict. Our results also revealed that vCA1-NAc shell inhibition impaired sociability and social novelty preference. Importantly, these effects were specific to conflict situations as they occurred independently of alterations in locomotion, anxiety, or cue avoidance/preference/novelty.

## Discussion

In the present study, chemogenetic inhibition of the glutamatergic projections of the vCA1 to NAc resulted in a decision-making deficit biasing animals towards avoidance in the face of motivational conflict, as evidenced by an increased time spent in the hub, increased latency to enter in the conflict cued arm (in males only), and potentiated retreat and risk assessment behavior prior to entering the conflict cued or neutral cued arm. We further demonstrated that these effects occur specifically under conditions of cue-elicited conflict, as vCA1-NAc inhibition did not impact any indices of decision-making or approach/avoidance behaviors in the absence of conflict (in conditioned cue preference/avoidance tests). Moreover, the observed effects were independent of any alterations in novelty processing, locomotion, or anxiety. Interestingly, we found that vCA1-NAc-inhibited animals were impaired in social interaction and social novelty-seeking. Together, our results implicate the vCA1-NAc shell circuit in driving the decision to approach in scenarios that involve motivational conflict or social novelty, revealing a nuanced role for the circuit in supporting approach-avoidance decisions under specific conditions.

We hypothesized that inhibition of the vCA1-NAc shell pathway would result in increased avoidance behavior in the face of an approach-avoidance conflict, in line with previous findings showing that pharmacological inhibition of the caudal NAc core and the vCA1 resulted in an increased avoidance of the conflict cue, characterized by a significant reduction in time exploring the conflict arm versus the neutral arm without any alterations in hub time, in the same task [14,15,34]. While there was some evidence suggesting that vCA1-NAc shell-inhibited animals exhibited a conflict avoidant phenotype—demonstrated by less time spent exploring the conflict versus neutral arm relative to control groups, slowed first entry into the conflict cued arm in the males, and increased retreat behaviors overall—we did not observe diminished time exploring the conflict versus neutral arms within-subjects. Instead, additional traits indicative of a decision-making deficit emerged following vCA1-NAc inhibition, distinct from those previously observed with NAc core and vCA1 manipulations. vCA1-NAc shell inhibited animals exhibited significantly increased central hub time, which we interpret as an increase in deliberation time and/or risk assessment. This is in line with the central compartment in the elevated plus maze (EPM) being similarly regarded as an area of choice point, where animals exhibit vicarious trial and error behaviors, which include "stretch-attend postures" (synonymous with our "checking" measure) as an index of risk assessment [35]. Accordingly, we observed increased "checking" behavior in the vCA1-NAc shell inhibited animals prior to the first entry into an arm during the conflict test, coupled with greater hesitancy to make an initial entry into the conflict cued arm (in males only), altogether pointing to an avoidance decision bias. Importantly, vCA1-NAc shell inhibition failed to induce the same behavioral pattern during conditioned cue preference and avoidance tests, or novelty cue preference test, suggesting that the decision-making deficit was specific to the presence of a conflict scenario. Thus, the role of NAc shell-projecting vCA1 neurons in cued approach-avoidance conflict processing is more complex than originally hypothesized, and our results point to a potential role of this pathway in the facilitation of approach during decision-making under motivational conflict. Indeed, this role is consistent with previous reports of elevated NAc shell dopamine being linked to rats that were classified as risk-taking, consistently choosing a larger reward paired with high probability of concomitant shock delivery over a certain, small reward [36] as well as NAc shell inhibition inducing risk-aversion (reduction in choice of larger reward with lower probability of delivery) in a probabilistic risk discounting task [37]. Interestingly, the human NAc shell exhibits increased activity only when participants choose to accept (aka approach) an offer with a 50/50 chance of gaining or losing money in a monetary

gambling task [38]. Altogether, vCA1 afferents to NAc shell may be important in biasing decisions towards "approach" in times of reward uncertainty and conflict.

In accord with previous findings, we observed impaired social discrimination in animals with vCA1-NAc shell circuit inhibition, evidenced by the failure to exhibit a preference for a novel over familiar rat or absence of a social stimulus. Previous work has implicated the vCA1 and its projections to the NAc in social memory [32,39,40]. However, in light of the current findings that vCA1-NAc inhibited rats exhibited a lack of preference for social interaction over a non-social stimulus in the sociability test, an alternative interpretation to the prevailing mnemonic explanation for these social discrimination impairments may be that vCA1-NAc inhibition diminishes the drive to seek the novel conspecific, which may be perceived as a conflict stimulus with inherent rewarding and aversive properties. Indeed, in the present study, we observed avoidance of the novel (over familiar) rat in the social memory test as well, similar to a previous finding in which an avoidance of a novel conspecific was reported following DA receptor antagonism of the NAc shell combined with cannabinoid agonism of the vHPC [41]. Interestingly, vCA1-NAc inhibition did not impact the detection of a novel set of bar cue stimulus in the Y-maze in the present study, suggesting that the presence of a motivational conflict involving the evaluation of the potential of reward versus punishment may be a prerequisite of vCA1-shell engagement. Relatedly, despite the abundance of data showing that vHPC lesions induce anxiolytic effects [42–45], we did not observe alterations in the performance of EPM or open field tests following vCA1-shell inhibition. These ethological tests of anxiety represent different forms of approach-avoidance conflict scenarios wherein animals must decide between approaching safe, enclosed spaces, or venturing into open areas that could potentially be dangerous. Of further note is that our findings are consistent with previous reports showing that vCA1-NAc shell circuit manipulations fail to perturb anxiety-related behavior, as measured by ethological tests [28,31].

The vHPC-shell pathway has been previously implicated in promoting stress-induced negative affective (depressive-like) states such as anhedonia and learned helplessness [28], susceptibility to chronic social stress, and inhibition of ethanol-seeking [46], suggesting that stress- or alcohol-induced maladaptive behavior may be mediated by an aberrant engagement of this pathway (but see [26] for evidence of inhibition of food seeking in stress-naïve animals). However, the vHPC-shell pathway has also been shown to support positively motivated behavior. For instance, LTP induction in the pathway promotes conditioned place preference in stress- and drug-naïve animals [33]. Similarly, optogenetic stimulation of the vHPC-shell circuit induces conditioned place preference and supports instrumental responding [47]. Furthermore, as discussed above, vHPC-shell activation promotes pro-social behavior [32,39,40]. Thus, the vHPC-shell pathway can mediate both negatively and positively motivated behavior under different circumstances, and the present findings extend this body of work by demonstrating that under situations in which stimuli of contrasting valences are presented simultaneously, the vCA1-shell pathway promotes approach decisions. The mechanism underlying this seemingly dichotomous function of the vHPC-shell circuit in promoting approach and avoidance behaviors under varying conditions is unclear. However, one possibility is that the nature of behavioral output (approach or avoid) is determined by the activation pattern of the NAc along the rostral-caudal axis, which is known to be functionally distinct, with the rostral aspect regulating appetitively motivated behaviors and caudal aspect aversively motivated behaviors [17–22,24]. Manipulations that reduce afferent input activity in the rostral NAc typically increase reward-seeking and consumption [29], whereas increase in activity in this region has the opposite effect of reducing food seeking. Calcium imaging evidence also points to divergent activity in rostral (decrease) versus caudal (increase) NAc-projecting vHPC neurons in mice making consummatory food port entries [29]. By extension, it may be stipulated that

in the present study, a reduction in vCA1 input led to an increase in avoidance bias via caudal shell mediated mechanisms, consistent with our previous work in which a pharmacological reduction in caudal NAc core activity led to an avoidance bias in the face of motivational conflict. vCA1/subiculum inputs extend throughout the rostro-caudal axis of the NAc shell, and these projection neurons are not arranged topographically within the vHPC [29], hence, future work is needed to address how differential afferent activation of the shell along the rostrocaudal axis is achieved to govern behavioral output.

The vCA1 has extensive extrinsic connectivity, including downstream targets such as the medial prefrontal cortex, amygdala, lateral septum, and lateral hypothalamus, as well as the NAc [18]. In the present study, we observed strong viral expression in other subregions of the ventral hippocampus (vDG, vCA3), and the lateral septum (LS) overlying the NAc shell, raising the possibility that the observed effects may have inadvertently resulted from targeting of the vCA3, or vCA1- lateral septum (LS) circuits (note that vDG is not known to have extrinsic connectivity beyond the medial temporal lobe). However, we find this unlikely for the following reasons. Firstly, as indicated by our CNO spread results (S1 Fig), the use of the direct intra-region microinfusion ensured that the targeting was specific and confined to NAc terminals. Secondly, a previous study from our laboratory selectively targeted the vCA3-LS and vCA1-LS pathways in the same task and found that their inhibition induced the opposite effect, potentiating approach behavior in the face of AAC [17]. The divergent vCA1 projections to subcortical regions such as the NAc and LS likely confer functional flexibility to vCA1, enabling the modulation of AAC across diverse conditions, and governing behavioral responses in dynamic and context-dependent ways.

Finally, while vCA1-NAc shell manipulations did not result in many sex differences, we did observe one drug treatment effect in males but not females, with inhibited males showing an increased latency to enter the conflict arm. In the few instances where sex differences were present in the hM4Di-expressing animals, regardless of vCA1-NAc inhibition, females showed fewer retreats overall during the conflict test and entered the conflict-cued arm less frequently than males. However, these effects were not replicated in the control GFP groups. These subtle sex differences may reflect the fact that females exhibit a different repertoire of defensive responses. Females have been shown to engage in more exploratory or "darting" behavior during and after exposure to a predatory or threat stimulus relative to males that exhibit more freezing behavior [48,49]. Future studies are warranted to further explore the divergence in the expression of conflict in males and females, and the underlying neural mechanisms driving these sex-specific responses.

In summary, these results demonstrate that the vCA1-NAc shell circuit is critically involved in biasing decision-making towards approach under situations of motivational and social ambivalence, distinct from other circuits such as the vCA3 or vCA1-lateral septal circuits that promote avoidance behavior under conflict [17]. Dysregulation of vCA1-NAc shell could underlie neuropsychiatric diseases by eliciting decision uncertainty which results in promoting approach in substance use disorders, while potentiating avoidance in mood disorders such as anxiety and depression [3,4,6,7,50]. Future research should be directed at further investigating the involvement of the vCA1-NAc shell pathway in contexts such as approach-approach or avoidance-avoidance conflict, which may reveal additional insights into the broader role of this circuit in decision-making under conflict.

## Materials and methods

### Animals

A total of 68 adult male and female Long-Evans rats (Charles River Laboratories, Quebec) weighing 225 to 325 g at the time of surgery were used in this study. Rats were pair-housed in

a colony room maintained at a constant temperature (21°C) and humidity (35% to 60%) with a 12-h light-dark cycle (light on at 7 AM). All behavioral testing, the order of which was counterbalanced between groups, occurred during the light hours and water was available ad libitum. Animals were food restricted for behavioral testing and were maintained at 85% to 90% of their pre-testing weight. All procedures were conducted in accordance with the regulations on animal experimentation established by the Canadian Council on Animal Care (CCAC) and approved by the Local Animal Care Committee at the University of Toronto (AUP 20012479).

## Surgery

Rats were anesthetized with isoflurane gas (Benson Medical, ON, Canada) and placed in a stereotaxic frame (Stoelting Co, Illinois, United States of America). A midline incision was made, and the fascia retracted to reveal bregma. Rats were then micro-injected bilaterally with 0.5 μl of either the AAV8-CAMKII-hM4Di-mCherry (Addgene, MA, TIter: $7 \times 10^{12}$ vg/ml) or AAV8-CAMKII-EGFP control viruses (Addgene, MA, TIter: $1 \times 10^{13}$ vg/ml) into the vCA1 (AP: −5.6, ML: ±5.8, DV: −7) over 5 min. After the viral injection, the injector remained in place for a further 10 min for the AAV to diffuse away from the tip. Following AAV injection, bilateral cannulae (26 gauge; Plastics One, Roanoke, Virginia, USA) were implanted into the NAc shell (AP: +1, ML: ±1, DV: −6.5) with the tips of the cannulae lying 1 mm above the target site. The guide cannulae were secured to the skull with 4 to 5 jeweler's screws embedded within dental cement. To maintain patency, stainless steel dummy cannulae were inserted into the guide cannulae. Rats were given, at minimum, 1-week post-surgery to recover, and the commencement of behavioral training was timed to ensure that the hM4Di receptors would have at least 6 weeks to express before activation.

## Drugs and microinfusions

Clozapine N-oxide (CNO-dihydrochloride, Hello Bio, United Kingdom), an agonist for hM4Di, was mixed with physiological saline (0.9%) to achieve a concentration of 1 mM. Prior to the first drug infusion, animals were acclimated to the infusion procedure. To achieve this, animals were sequentially exposed to the room, the restraint hold and the removal and insertion of the dummy cannula. On infusion test days, rats were bilaterally infused with 0.3 μl of either vehicle saline or CNO at an infusion rate of 0.3 μl/min via 30-gauge microinjectors connected to a 10 μl Hamilton syringe placed on an infusion pump (Harvard Apparatus, MA). Post-infusion, the infusers were left in place for an additional minute to allow for drug diffusion. Animals were then given a 10-min adjustment period prior to behavioral testing.

## Overview of behavioral procedures

Animals were operated and tested successively in cohorts of 6 to 12 animals (roughly equal numbers of hM4Di CNO and SAL or GFP CNO and SAL). Following recovery from surgery, animals were administered a series of behavioral tests as shown in the timeline at the top of Figs 2–4A. Unless otherwise stated CNO/SAL microinfusions were conducted 10 to 15 min prior to each behavioral test session, and the maximum number of repeated intra-cerebral (IC) infusions given to the animals was 7, consistent with other reported IC studies from our laboratory. All tests with CNO/SAL microinfusion were separated by at least 48 h to ensure sufficient time for drug washout to occur.

All animals began their training in the mixed valence Y-maze task first. Only those animals that successfully acquired the cue-outcome associations (all but two, 1 hM4Di CNO female, 1 hM4Di SAL female) were administered a cued conflict test (hM4Di Sal: $n = 17$ (6 females), hM4Di CNO: $n = 17$ (7 females), GFP Sal: $n = 15$ (7 females), GFP CNO: $n = 18$ (8 females)).

Following the completion of the conflict test, animals received a refresher conditioning session, and the following day were administered the conditioned cue preference and conditioned cue avoidance tests in counterbalanced order, with the first test beginning 10 min following drug or saline infusion. Three male animals were eliminated from the study after the conflict test due to blocked cannulae ($n = 2$) or illness ($n = 1$) with the following animals remaining in the study: hM4Di Sal: $n = 16$ (6 females), hM4Di CNO: $n = 16$ (7 females), GFP Sal: $n = 15$ (7 females), GFP CNO: $n = 17$ (8 females)). In the last 2 cohorts of animals (hM4Di CNO: $n = 8$ (4 females), GFP CNO: $n = 8$ (4 females), an additional novelty cue preference test was administered after a refresher conditioning session. All animals except the last cohort ($n = 6$) were then subjected to the sociability and social memory tests. Data from 8 animals were subsequently excluded from the sociability and social memory data analysis due a change in the method of testing (see details later) such that the number of animals included in the data analysis were: hM4Di Sal: $n = 15$ (7 females), hM4Di CNO: $n = 11$ (7 females), GFP Sal: $n = 12$ (6 females), GFP CNO: $n = 14$ (7 females). Thereafter, all animals including the 2 animals whose data were excluded from the Y-maze experiment due to non-learning (hM4Di Sal: $n = 17$ (7 females), hM4Di CNO: $n = 17$ (8 females), GFP Sal: $n = 15$ (7 females), GFP CNO: $n = 17$ (8 females)) underwent the locomotor and then the elevated plus maze. Finally, animals received their final infusion of CNO or saline, and all but 3 animals in the last cohort (GFP saline male and female, 1 GFP CNO male) were administered the open field test and sacrificed thereafter.

## Mixed valence Y-Maze task

**Apparatus.** Behavioral training and testing occurred in a six-arm radial maze apparatus (Med Associates, USA). Access to the arms (45.7 cm (L), 9 cm (W), 16.5 cm (H)) was controlled by automated guillotine-like doors. Only 3 of the 6 arms (forming a Y-maze) were used at any one time. Each arm was enclosed with plexiglass walls, a removable lid and contained a steel grid floor. The walls and lid were covered with red cellophane to obscure extra-maze stimuli and the steel grids were connected to a foot shock generating device (Med Associates, USA). Each arm also contained a receding well at the end, which was connected to a syringe via polyethylene tubing to allow the delivery of a liquid sucrose solution (20%). The maze was wiped down with 70% ethanol solution after each training session to eliminate odor traces, and rotated by varying degrees (60˚, 120˚, or 180˚) at the end of each day to minimize conditioning to intra-maze cues. A ceiling-mounted camera was used to record test sessions.

## Cues

During training and testing, 3 sets of visuotactile bar cues with different textures, colors, and reflective properties (45 cm × 2.5 cm) were attached to the lower portion of the maze arms. Each bar insert became associated with either sucrose delivery (appetitive), mild foot shock administration (aversive), or no outcome (neutral). The valence of cues for each rat was determined following the unvalenced cue habituation session (described below). This allowed us to counterbalance any innate preferences for a cue by assigning the opposite valence during conditioning.

## Habituation

As described in detail previously [14,15,17,45], rats received 4 separate 5-min habituation sessions. During the first session, rats were habituated to the Y-maze apparatus without the insertion of any bar cues. The rats were first placed in the central hub. After a minute, the doors leading to all 3 arms were raised and the rats were allowed to freely explore the maze for 5 min. During the second session, the visual-tactile cues were inserted into the arms. Rats then freely

explored the maze and cues for 5 min. Their time spent in each arm, each with a different cue, was recorded and used to assign cue valence. Specifically, the least preferred cue was assigned as the appetitive cue, the most preferred cue as the aversive cue, and the remaining cue as the neutral cue. During the third session, rats were presented with 2 sets of cues in 2 arms. One arm contained a combination of the to-be-assigned appetitive and aversive cue while the other contained the neutral cue. This ensured that the combination of these 2 cues, which occurs during the final conflict test, was not a novel stimulus. During the final habituation session, rats were placed into the central hub for 30 s. They were then confined in each of the 3 un-cued arms, separately, for 2 min before being returned to the hub.

## Cue conditioning

Animals underwent 8 to 12 conditioning sessions to acquire appetitive and aversive cue conditioning. Each session began with 30 s in the hub, followed by 2 min in each of the individual arms. In the arm containing the appetitive cue, rats received $4 \times 0.4$ ml, 20% sucrose delivered at random intervals averaging 30 s. In the arm containing the aversive cue, rats received 4 mild foot shocks (1 s, 0.25 to 0.32 mA) delivered at random intervals. In the arm with the neutral cue, no outcomes were given. The order of arm presentation was varied daily to prevent animals associating the outcomes with the order of presentation. Additionally, arm cue placement was counterbalanced between animals and varied between sessions. The maze was also rotated daily to limit the effect of any extra-maze stimuli.

## Conditioned cue acquisition test

After every fourth conditioning session, rats underwent a conditioned cue preference/avoidance test to assess their learning. Rats freely explored the cues in the maze for 5 min in the absence of sucrose or shock. Time spent in each arm was measured. Successful acquisition of conditioned cue preference and avoidance was indicated by rats spending more time in the appetitive arm than the neutral and aversive arms (conditioned cue preference) and spending the least amount of time in the aversive arm (conditioned cue avoidance).

## Mixed valence conflict test

Before being placed into the radial maze for the mixed valence conflict test, rats received either a CNO or saline microinfusion, as described above. During this test, rats freely explored 2 arms for 5 min. One arm contained both the appetitive and aversive cues while the other arm contained only the neutral cue. We recorded multiple behavioral measures as indices of approach/avoidance, including the time spent in each cued arm and the hub, latency to enter each arm for the first time, the number of full body entries in to the arms, retreats (half body entries into the arm followed by rapid withdrawal into the hub, or back treading once inside the arm) and risk assessment behavior (head was tracked to quantify the number of times the rats spent looking/checking back and forth between arms before making their first entry into an arm).

## Cue preference and avoidance tests

Following the conflict test, animals underwent a refresher cue conditioning session. A day later, rats received a drug (CNO or saline) microinfusion as described above, and underwent cue preference and cue avoidance tests, the order of which were counterbalanced across animals (and conducted a minimum of 30 min apart). During these tests, rats freely explored two of the arms for 5 min in extinction. One arm contained either the appetitive or the aversive

cue while the other arm contained only the neutral cue. Total time spent in each arm, hub, as well as other indices of approach/avoidance behaviors were measured.

### Cue novelty test

A subgroup of animals ($n$ = 16) received another refresher cue conditioning session and were then administered a cue novelty test prior to which they received a CNO microinfusion in a between-subject design as the novelty test cannot be repeated ($n$ = 8 hM4Di, $n$ = 8 GFP). In this test, animals were allowed to freely explore 2 arms of the Y-maze for 5 min, this time with 1 arm containing a novel set of cues, and another containing the neutral cues. Total time spent in each arm and hub, as well as other indices of approach/avoidance behaviors were measured.

### Social preference and memory test

We used a social discrimination paradigm to test the effects of vCA1-NAc shell inactivation on social preference and memory. Rats were habituated to a circular open arena [136 cm (D), 75 cm (H)] with 2 empty wire cages [25 cm (L) × 20 cm (W) × 15 cm (H)] placed 50 cm apart for 15 min. Following the habituation session, rats received a microinfusion of either saline or CNO, and were then placed into the center of the arena to explore the apparatus and allowed to interact with a caged conspecific and an empty cage for 10 min before being removed. After a 20-min delay period, rats were placed back into the area, but this time with 2 caged conspecifics to interact with (1 novel and 1 familiar). In a small subset of animals ($n$ = 8), the rats were presented with 2 caged conspecifics (instead of one) in the first phase. However, this version was soon abandoned for the remaining animals as we wished to utilize the first phase as a sociability test (preference for social stimulus over no stimulus). Thus, only data from the remaining animals are reported. All tests were recorded using a video camera and the time spent interacting with and sniffing each cage was measured.

### Locomotor test (LM)

After a saline or CNO microinfusion, all animals were placed into individual cages (44 cm [L] × 24 cm [W] × 20 cm [H]) with standard bedding material and stainless-steel cage lids. Animals were left for 1 h while their locomotor behavior was monitored using an overhead camera and processed by ANY-maze tracking software. Total distance traveled (in m) was recorded in 5-min time bins.

### Elevated plus maze (EPM) test

An elevated plus maze task was used to measure unconditioned anxiety levels. The apparatus was placed in a novel room and was elevated 50 cm from the floor. It contained a central hub [10 cm (L) × 10 cm (W)] that connected 4 arms [40 cm (L) × 10 cm (W)], two of which were enclosed by walls [22 cm (H)]. After infusion of either saline or CNO, rats were placed in the central hub of the maze facing an open arm and given 10 min to explore the maze. Time spent in the open and closed arms as well as the middle area were measured. Additionally, the number of entries into each arm was recorded.

### Open field test

An open field test was conducted as a final test prior to euthanasia, to endogenously activate c-Fos labeling in the region of interest for immunohistochemical analysis. Rats were placed in an open field arena [45 cm (L) × 45 cm (W) × 40 cm (H)] for 10 min. This apparatus consisted of 4 black opaque walls and the floors were lined with fresh bedding. The arena was divided into

a $3 \times 3$ grid, and the time spent in the center zone (grid in the center, $15 \times 15$ cm) and periphery of the apparatus was analyzed using NOLDUS EthoVision XT software.

## Histology

Rats underwent transcardial perfusions with 4% paraformaldehyde (PFA) within 75 to 90 min after the completion of the open field test, and their brains were extracted. A small sample of GFP-expressing control animals ($n = 3$) underwent a microinfusion of CNO (1 mM) and rhodamine 6G (dissolved in saline at a concentration of 0.02%, Fisher Scientific, Canada), and were sacrificed 10 to 15 min later for the extraction of their brains. The brains were sliced into 50 μm coronal sections using a vibratome (Leica VT1200S). A subset of these slices was then analyzed to confirm viral/fluorophore expression and cannula placement.

## c-Fos immunohistochemistry

Brain sections were treated with 1% hydrogen peroxide, then incubated in 0.5% 1,3,5-Trinitrobenzene (TNB) blocking solution for 1 h at room temperature and incubated overnight at 4°C with rabbit c-fos antibody (1:5,000 dilution, Synaptic Systems, Goettingen, Germany). Following incubation with the primary antibody, sections were washed in PBS ($5 \times 5$ min) and incubated overnight with secondary antibody (peroxidase conjugated donkey-anti-rabbit 1:250, Jackson Immunoresearch, Baltimore, Pennsylvania, USA). The Tyramide Signal Amplification procedure was then employed, using NHS-fluorescein for the hM4Di-expressing brains, or NHS Rhodamine for the GFP control brains (Thermo Fisher Scientific, MA) as dye substrates. After a final round of PBS washes, brain slices were mounted on gelatin-coated slides and air dried before being coverslipped with Fluoroshield Mounting medium with DAPI for nuclear staining (Abcam, MA).

## Cell imaging and counting

hM4Di and GFP expression, and c-Fos immunoreactivity were subsequently visualized at 4×, and 20× magnification using the NIKON Ti2-E microscope (NIKON, NY). GFP expressing cells and c-Fos positive cells conjugated with TSA-fluorescein were visualized using the FITC filter (excitation: 467–498 nm; emission: 513–556 nm), while hM4Di-mCherry expressing cells and c-Fos positive cells conjugated with TSA-Rhodamine were visualized using the TexRed filter (excitation: 532–587 nm; emission: 608–683 nm). Quantification of c-Fos positive cells was achieved using 2 images of coronal sections of the NAc taken at 4× magnification from each animal. Projections from the ventral CA1 to the shell were demarcated as the regions of interest, and the number of c-Fos-positive cells within the NAc shell and core were counted by converting the images into 8-bit and using ImageJ software (Rasband, W.S., US National Institutes of Health).

## Statistical analysis

All data collected were analyzed using SPSS (v25) and graphed in GraphPad Prism version 10.0.2 (GraphPad Software, La Jolla, California, USA). We employed a mixed-model design, assigning animals to one of 4 between-subject groups based on construct (hM4Di or GFP [empty vector control]) and drug (CNO or saline), with sex as an additional between-subjects factor and various dependent measures as within-subject factors (as described below). To focus on key comparisons and facilitate the interpretation of results, we analyzed most data from each construct group (hM4Di versus GFP) separately, with CNO versus SAL serving as the control comparison within each group, unless otherwise stated. The GFP CNO group

served as a control for potential off-target effects of CNO, and the saline controls served as a baseline to account for any construct-related effects independent of CNO administration. Male and female data were graphically separated in instances where significant sex differences were observed.

To assess the acquisition of cue-outcome associations during the training period, acquisition data (time spent in each arm) were subjected to a repeated measures analysis of variance (ANOVA) test with Arm (Appetitive, Neutral, Aversive) as a within-subjects factor. To determine whether there were any preexisting group differences, Sex (Male, Female), and future Drug Condition (Saline, CNO) were used as between-subjects factors.

Data from the Y-maze conflict test and cue preference/avoidance tests (time spent in each arm, latency to enter, "risk assessment" behavior, number of entries and retreats) were subjected to a repeated measures ANOVA with Sex and Drug as between-subjects factors and Cue (Conflict, Neutral; Appetitive, Neutral; Aversive, Neutral) as a within-subjects factor. Additionally, we calculated a difference score (time spent in the conflict cued arm minus time spent in the neutral arm) to assess the degree of preference for the conflict arm, and this variable was subjected to an ANOVA with Sex and Drug as between-subjects factors.

A decision-making composite z-score was computed as a measure that combines multiple behavioral indicators (hub time, "checking" behavior, and latency to enter the conflict arm) to reflect the level of deliberation or hesitancy during decision-making. Each of these individual measures was first standardized by calculating their z-scores across sex, drug, and virus groups. The 3 z-scores were then averaged to create a single composite score, and subjected to ANOVA with Virus, Sex, and Drug as between-subjects factors. Higher composite z-scores indicate more time spent deliberating or hesitating, which suggests increased decision-making difficulty or caution.

Data from the novelty cue preference were also subjected to ANOVA with Virus as a between-subject factor and Cue (Novel, Neutral) as a within-subject factor. For all tests, central hub time was analyzed separately using ANOVA with Drug and Sex as between-subject factors.

Social data (interaction time) were analyzed with Drug and Sex as between-subjects factors and Stimulus (Novel versus no conspecific for first social preference test) or Novelty (Familiar, Novel for social memory test) as a within-subjects factor.

Locomotor data (distance traveled) were split into twelve 5-min time bins and analyzed using a repeated measures ANOVA with Sex and Drug as between-subjects factors and Bin [1–12] as a within-subjects factor. Similarly, both anxiety tests (elevated plus maze and open field) were subjected to repeated measures ANOVA. For EPM, time spent in each arm was analyzed with Drug and Sex as between-subjects factors and Arm (Open, Closed) as a within-subjects factor. For open field test data, time spent in the central grid was analyzed by ANOVA with Drug and Sex as between-subjects factors.

Finally, the density of c-Fos positive cells in the NAc shell and core of the hM4Di virus group was analyzed using a 3-way ANOVA with Drug and Sex as between-subjects factors.

All tests were accompanied by tests of homogeneity of variance and sphericity, and significant main/interaction effects were followed up with planned comparisons and post hoc tests where appropriate, using Bonferroni correction to control for Type 1 error in multiple comparisons. The alpha level was set at $p < 0.05$, and Bonferroni corrected $p$-values are reported.

## Supporting information

**S1 Fig. Intra-NAc shell infusions of CNO-rhodamine in a GFP expressing rat.** Images (A–F) represent coronal sections along the anteroposterior (AP) axis (approx. +2.1 to +1.0),

showing GFP-expressing fibers from the ventral hippocampus (vHPC) terminating in the NAc shell. Bilateral rhodamine-laced CNO infusions (0.3μ l) were localized to the NAc shell and their spread was found to be co-localized within GFP-expressing areas. No encroachment of CNO-rhodamine into overlying areas (e.g., septum) was observed. Scale bar (applicable to all images) represents 1,000 μm.
(JPG)

**S2 Fig. Intra-NAc CNO or Saline (SAL) infusions had no effect on approach-avoidance conflict in GFP-expressing (control) animals.** (**A**) After 2 to 3 rounds of 4 conditioning sessions, 33 animals (GFP Sal: $n = 15$ (7 females), GFP CNO: $n = 18$ (8 females), symbols with red borders represent female data) successfully acquired cue-outcome associations, demonstrating a greater amount of time in the appetitive vs. the neutral arm ($p < 0.001$) and less time in the aversive vs. the neutral arm ($p < 0.001$). (**B**) In the conflict test, the time spent in the conflict or neutral cued arms was not significantly different between the CNO or SAL-infused groups, nor between males and females. (**C**) Neither group exhibited a preference for the conflict or neutral-cued arm. (**D**) Amount of time spent in the hub (central compartment); (**E**) Risk assessment (checking) behavior; (**F**) Latency to enter the conflict or neutral cue arms for the first time; (**G**) Number of retreats; (**H**) Number of arm entries were not significantly different between the CNO or SAL-infused groups, or between males and females. Data show means ± SEM. The data underlying this figure are available at: https://osf.io/pwcer/.
(TIFF)

**S3 Fig. Intra-NAc CNO or Saline (SAL) infusions had no effect on conditioned cue preference and avoidance tests in GFP-expressing (control) animals.** (**A, B**) GFP-expressing animals (GFP Sal: $n = 15$ (7 females), GFP CNO: $n = 18$ (8 females), symbols with red borders represent female data) spent significantly longer time in, and made more entries into, the appetitive vs. neutral arm in the cue preference test, irrespective of drug treatment. (**C**) No sex or drug treatment difference in latency to enter the appetitive vs. neutral arms was observed. (**D, E**) There was no difference in the hub time or risk assessment during the cue preference test between any groups. (**F**) GFP-expressing animals spent less time in the aversive vs. neutral arm in the cue avoidance test. (**G, H**) Latency to enter the aversive cued arm for the first time was significantly higher compared to latency to enter the neutral arm in both groups, irrespective of sex. (**H**) Less entries into the aversive arm over the neutral arm in the cue avoidance test were observed, irrespective of drug treatment or sex. (**I**) The number of retreats observed in the aversively cued arm was significantly higher than in the neutral cued arm. (**J, K**) There was no difference in hub time and risk assessment behavior between all groups in the cue avoidance test. Data show means ± SEM. * $p < 0.01$, ** $p < 0.01$, ***$p < 0.001$. The data underlying this figure are available at: https://osf.io/pwcer/
(TIFF)

**S4 Fig. Intra-NAc, CNO, or Saline (SAL) infusions had no effect on social interaction tests in GFP-expressing (control) animals.** (**A**) All GFP-expressing animals, irrespective of drug treatment or sex (GFP Sal: $n = 12$ (6 females), GFP CNO: $n = 14$ (7 females), symbols with red borders represent female data), spent significantly more time interacting with the novel conspecific over an empty cage, showing preference for social interaction. (**B**) All animals, irrespective of drug treatment or sex, spent more time interacting with the novel rat over the familiar rat, (**C**) exhibiting a preference for the novel rat. Data show means ± SEM. ***$p < 0.001$. The data underlying this figure are available at: https://osf.io/pwcer/.
(TIFF)

**S5 Fig. Intra-NAc CNO or Saline (SAL) infusions had no effect on locomotor activity or innate anxiety in hM4Di- or GFP-expressing (control) animals.** (**A**) Timeline of final set of experimental procedures involving locomotor activity test, elevated plus maze, and finally, open field test. Animals (hM4Di Sal: $n = 17$ (7 females), hM4Di CNO: $n = 17$ (8 females), symbols with red border depict female data) were microinfused with CNO or Saline prior to the locomotor test, and open field test. (**B**) Females showed elevated locomotor activity overall compared to males, but this was irrespective of vCA1-NAc shell inhibition. (**C**) All animals spent significantly more time exploring the closed vs. open arms of the elevated plus maze with no group (sex or drug treatment) differences. (**D, E**) All groups spent equal times exploring the centers of the EPM and open field. (**F**) The microinfusion of CNO in GFP-expressing animals males and females did not affect locomotor activity (GFP Sal: $n = 15$ (7 females), GFP CNO: $n = 17$ (8 females)). (**G**) All animals spent significantly more time exploring the closed vs. open arms of the elevated plus maze with no group differences. (**H, I**) CNO microinfusions did not alter the time spent in the center of the EPM or Open field apparatus. Data show means ± SEM. The data underlying this figure are available at: https://osf.io/pwcer/. (TIFF)

## Author Contributions

**Conceptualization:** Rutsuko Ito.

**Formal analysis:** Dylan Patterson, Rutsuko Ito.

**Funding acquisition:** Andy C. H. Lee, Rutsuko Ito.

**Investigation:** Dylan Patterson, Nisma Khan, Emily A. Collins, Norman R. Stewart, Kian Sassaninejad, Dylan Yeates.

**Methodology:** Rutsuko Ito.

**Project administration:** Rutsuko Ito.

**Supervision:** Rutsuko Ito.

**Visualization:** Rutsuko Ito.

**Writing – original draft:** Dylan Patterson, Rutsuko Ito.

**Writing – review & editing:** Nisma Khan, Emily A. Collins, Norman R. Stewart, Andy C. H. Lee.

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
