## [Editor Report · Decision Letter 0]

25 Jun 2024

Dear Rutsuko, 

Thank you for submitting your manuscript entitled "The ventral hippocampal-nucleus accumbens shell circuit drives approach decisions under social novelty and learned cue approach-avoidance conflict", together with a revision plan in response to reviews from Biological Psychiatry, for consideration as a Short Report by PLOS Biology.

I have now had a chance to discuss your manuscript and revision plan with an Academic Editor with relevant expertise, and I am writing to let you know that we would like to invite you to complete your revision. Once your revision is complete, we would then aim to re-review the manuscript with the original reviewers from Biological Psychiatry - or, if that is not possible, we will enlist the help of 1-2 arbitrating reviewers. 

**Please note**

1) Before we can formally invite a revision for your study, we need you to complete your submission by providing the metadata that is required for full assessment. To this end, please login to Editorial Manager where you will find the paper in the 'Submissions Needing Revisions' folder on your homepage. Please click 'Revise Submission' from the Action Links and complete all additional questions in the submission questionnaire.

Once your full submission is complete, your paper will undergo a series of checks. After your manuscript has passed the checks I will send you another decision letter, where I formally put your manuscript 'in revision' and pass along a few editorial comments on the revision plan. To provide the metadata for your submission, please Login to Editorial Manager (https://www.editorialmanager.com/pbiology) within two working days, i.e. by Jun 27 2024 11:59PM.

2) I did reach out to the editors at Biological Psychiatry to see if they could transfer the peer review history for your manuscript to us - and they said they are happy to do so, but can only do that if the corresponding author of the manuscript contacts them first. I therefore ask that you reach out to the editors at BP and request that they share the peer review history for your manuscript with us. As I may go out on parental leave any day now, please give them the general email address for our editorial offices: "biology_editors@plos.org", and ask that they forward the relevant details there. 

Kind regards,

Luke

Lucas Smith, Ph.D.

Senior Editor

PLOS Biology

lsmith@plos.org

---

## [Editor Report · Decision Letter 1]

26 Jun 2024

Dear Rutsuko,

Thank you again for submitting your manuscript "The ventral hippocampal-nucleus accumbens shell circuit drives approach decisions under social novelty and learned cue approach-avoidance conflict", with reviews from Biological Psychiatry, for consideration at PLOS Biology, and thanks for completing the metadata. 

As mentioned in my last email, after discussing your manuscript and revision plan with an Academic Editor with relevant expertise, we would like to invite you to complete the revision ~as detailed in your revision plan. I would emphasize that we think it will be important to address the the concerns regarding structure and readability of the manuscript, in addition to thoroughly addressing the technical concerns and questions raised in the previous reviews. 

As a last note (I included this in my last email, as well but just re-iterating it here): At your earliest convenience, we ask that you reach out to the editors at Biological Psychiatry and request that they share the peer review history for your manuscript with us, as we will need that confirmation before we re-review your study. As I may go out on parental leave any day now, please give them the general email address for our editorial offices: "biology_editors@plos.org", and ask that they forward the relevant details there.

Given the extent of revision needed, we cannot make a decision about publication until we have seen the revised manuscript and your response to the reviewers' comments. Once your revision is complete, we will aim to re-review the manuscript with the original reviewers from Biological Psychiatry - or, if that is not possible, we will enlist the help of new arbitrating reviewers.

**IMPORTANT - SUBMITTING YOUR REVISION**

*Re-submission Checklist*

*Published Peer Review*

*PLOS Data Policy*

*Blot and Gel Data Policy*

Sincerely,

Luke

Lucas Smith, Ph.D.

Senior Editor

PLOS Biology

lsmith@plos.org

---

## [Decision Letter · Decision Letter 2]

12 Dec 2024

Dear Rutsuko, 

Thank you for your patience while we considered your revised manuscript "The ventral hippocampal-nucleus accumbens shell circuit drives approach decisions under social novelty and learned cue approach-avoidance conflict" at PLOS Biology, which was submitted with portable peer reviews from Biological Psychiatry. Your revised study has now been evaluated by the PLOS Biology editors, the Academic Editor and by two of the original reviewers. 

As you will see in the comments below, the reviewers agree that the study has been strengthened, but reviewer 3 has a few lingering concerns. Having discussed these within the team and with the Academic Editor, we think that these can largely be addressed with textual changes and we would not require that you add additional controls as suggested in point 2. (While we think those controls would be nice, we think the data are strong enough without them). We would, however, encourage you to add, as a supplemental figure, the validation data that reviewer 3 comments on in point 3 and to address the other concerns raised. 

In light of the reviews, which you will find at the end of this email, we are pleased to offer you the opportunity to address these remaining points in another revision that we anticipate should not take you very long. We will then assess your revised manuscript and your response to the reviewers' comments with our Academic Editor aiming to avoid further rounds of peer-review, although might need to consult with the reviewers, depending on the nature of the revisions.

**IMPORTANT: In addition to addressing the last reviewer comments, please also attend to the following editorial and data related requests. 

1) TITLE: While we think the current title does a good job capturing the message of the study, I have been wondering if this might be streamlined a bit to make it more accessible to our broad audience. As an example, if you agree (and feel this is supported), we would suggest you change the title to something like: 

"A neural circuit between the ventral hippocampus and nucleus accumbens regulates approach decisions during motivational conflict"

I am happy for you to refine further

2) FIGURES: Sorry for only raising this now, but please note that PLOS Biology Short Reports can have a maximum of 4 figures, while the current work has 5. I therefore request that you streamline the figures a bit to meet this requirement. I suspect the easiest way to condense the data into 4 figures would be to just combine 2 of the existing figures? Feel free to reach out to us if you would like to discuss this further. 

3) ETHICS STATEMENT: Thank you for providing an ethics statement in your methods section. Please update this to include the approval number for the protocol, approved by the IACUC at University of Toronto. 

4) DATA: Thank you for depositing the underlying data related to your study to OSF. For the most part this looks good to me but can you please either update the file on OSF, or add a readme that more clearly spells out which figures the data relates to? Please also add a sentence to each figure legend pointing readers to this file. (For example, you can say "the data underlying this figure can be found at https://osf.io/pwcer/?view_only=46274147a9214b6a994322b90f67d378") 

5) CODE: Per journal policy, if you have generated any custom code during the course of this investigation, please make it available without restrictions. Please ensure that the code is sufficiently well documented and reusable, and that your Data Statement in the Editorial Manager submission system accurately describes where your code can be found.

**IMPORTANT - SUBMITTING YOUR REVISION**

*Resubmission Checklist*

*Published Peer Review*

Sincerely,

Luke

Lucas Smith, Ph.D.

Senior Editor

PLOS Biology

lsmith@plos.org

REVIEWS:

Reviewer #1: The authors have addressed my comments and I consider this paper now suitable for publication.

Reviewer #3: While I recognize the authors' effort to improve the paper, there are still few observations/limitations and additional comments to be considered and addressed to improve the manuscript in general (for this or other journal). 

1-Fig1S (related to Reviewer 1; Point 1)- the authors should just state that a limitation may include CNO spreading into the nearby regions (i.e. core and septum). it's also clear that vHPC/CA1 projects to the NAc core (based on the terminals (green labeling)). 

2- Figs2-3 and 5- Unclear: would it be better use DREADDS (hM4Di-mCherry) or empty vector (mCherry or GFP) groups both with CNO (as in Fig 1)? 

3- Related to Reviewer 3; Point 7- As the authors mentioned "Of note is that we validate the neural inhibition of targeted areas/circuits using cfos staining at the end of every experiment, and observe expected reduction in cfos activation after CNO infusions". Would be useful to show this in the supplement? 

4- Related to Reviewer 3; Point 16 (Fig4)- I can't still find the m/f labels figures/panels. 

5- Related to Reviewer 3; Point 23- what phase? (mix valence, acquisition or conflict).

---

## [Editor Report · Decision Letter 3]

20 Dec 2024

Dear Rutsuko,

Thank you for the submission of your revised Short Report "Ventral hippocampus to nucleus accumbens shell circuit regulates approach decisions during motivational conflict" for publication in PLOS Biology and thank you for addressing the last reviewer and editorial requests in this revision. On behalf of my colleagues and the Academic Editor, Thomas Klausberger, I am pleased to say that we can in principle accept your manuscript for publication, provided you address any remaining formatting and reporting issues. These will be detailed in an email you should receive within 2-3 business days from our colleagues in the journal operations team; no action is required from you until then. Please note that we will not be able to formally accept your manuscript and schedule it for publication until you have completed any requested changes.

**IMPORTANT: As you address any formatting requests to come, please also attend to the following editorial request: 

1) Thanks again for providing the underlying data related to your study on OSF. Please make sure to make this dataset public before publication and to generate a DOI for it, as this will create a permanent record. Once you have the DOI, please update the data availability statement in our electronic manager system, and update the figure legends to reference that. Here is a link where you can learn more about how to generate a DOI for this dataset: https://help.osf.io/article/220-create-doi

PRESS

Sincerely, 

Luke 

Lucas Smith, Ph.D.

Senior Editor

PLOS Biology

lsmith@plos.org